# Health beliefs and health seeking behavior towards lymphatic filariasis morbidity management and disability prevention services in Luangwa District, Zambia: Community and provider perspectives

**Patricia Maritim**[1]*, **Adam Silumbwe**[2], **Joseph Mumba Zulu**[1], **George Sichone**[3], **Charles Michelo**[4]

**1** Department of Health Promotion and Education, School of Public Health, University of Zambia, Lusaka, Zambia, **2** Department of Health Policy and Management, School of Public Health, University of Zambia, Lusaka, Zambia, **3** Participatory Research and Innovations Management (PRIM), Lusaka, Zambia, **4** Department of Epidemiology and Biostatistics, School of Public Health, University of Zambia Lusaka, Zambia

\* triciamarie20@gmail.com

**Data Availability Statement:** All relevant data is included within the manuscript. Participants in the

## Abstract

### Background

Morbidity management and disability prevention (MMDP) services are essential for the management of chronic stages of lymphatic filariasis (LF) infection. However, there is limited information on health beliefs and health seeking behavior towards MMDP services for LF in endemic regions of Zambia. This study sought to document health beliefs and health seeking behavior towards MMDP services for LF in Luangwa District, Zambia.

### Methods

This was an exploratory qualitative study conducted with community members including LF patients, community health workers and healthcare providers. Data was collected through a series of four focus group discussions stratified by sex and 26 in-depth interviews. Data was analyzed by thematic analysis using NVivo software.

### Results

The perceived causes of the chronic manifestations of LF included; contact with animal faeces, use of traditional herbal aphrodisiacs (*mutoto)*, witchcraft and sexual contact with women who were menstruating or had miscarried. LF patients opted to visit traditional healers before going to health facilities. Hydrocele patients were afraid of hydrocelectomies as they were thought to cause infertility or death. Very few community members were able to identify any home and facility-based care strategies for lymphoedema. Health system and cultural barriers to seeking healthcare included; long distances to the health facilities, lack of

study are drawn from a small community and the information provided in the manuscript if combined with that in the complete and linked data set could compromise their privacy. Specific data requests can be directed to wworthington@taskforce.org.

**Funding:** This study was funded by the Coalition for Operational Research on Neglected Tropical Diseases (COR-NTD),(https://www.ntdsupport.org/cor-ntd/ntd-connector/term/ntdsc), under Grant Number:NTD-SC #163D and was awarded to CM. The funders had no role in study design, data collection and analysis, decision to publish, or preparation of the manuscript.

**Competing interests:** The authors have declared that no competing interests exist.

awareness of existing MMDP services, perceived costs of accessing MMDP services, gender and social norms, and fear of stigmatization.

## Conclusion

Health seeking behavior for LF in the district is mainly driven by negative beliefs about the causes of the disease and lack of awareness of available MMDP services and homecare strategies. Lymphatic filariasis programs should promote strategies that seek to empower patients and community members with the required information to access and use the MMDP services at the health facilities, as well as adhere to self-care practices in their households.

### Author summary

Lymphatic filariasis (LF) infection if untreated results in fluid accumulation in the limbs or breasts (lymphedema) or genitalia (hydrocele) that is painful and causes great discomfort. Morbidity management and disability prevention (MMDP) strategies such as surgery for hydrocele, treatment of acute attacks and management of lymphedema are necessary for the management of the advanced stages of LF. However, very few countries including Zambia, have adequate information on the health beliefs and health seeking behavior of communities living in endemic areas towards MMDP services for LF. This study sought to explore community and health provider perspectives towards MMDP services for LF in a highly endemic region, Luangwa District, Zambia, between February and April 2019. Some of the perceived causes of lymphedema and hydrocele were; contact with animal faeces, use of traditional herbal aphrodisiacs (*mutoto*), witchcraft and sexual contact with women who were menstruating or had miscarried. There was limited knowledge of home-based and facility-based care strategies for lymphoedema. Nevertheless, patients would often go to health facilities after visiting traditional healers and observing no improvement. Barriers to accessing healthcare included; long distances to the health facilities, lack of awareness of existing MMDP services, perceived costs of accessing healthcare services, gender and social norms and fear of stigmatization.

## Introduction

Lymphatic filariasis (LF), a neglected tropical disease, causes permanent disability through chronic manifestations of lymphedema, elephantiasis and hydrocele accounting for 1.36 million disability adjusted life years [1]. Globally more than 858 million people living in 49 endemic countries face the risk of infection and an estimated 40 million have chronic manifestations of the disease [2]. Disease control programs have mostly targeted the interruption of LF transmission through mass drug administration (MDA) whilst placing less emphasis on promoting morbidity management and disability prevention services (MMDP) for those presenting with chronic manifestations. The basic care package of MMDP services includes individual treatment for episodes of adenolymphangitis (acute attacks), destruction of microfilaria, management of lymphedema to prevent disease progression, and surgery for hydrocele [3]. Since 2000, MDA programmes for LF programs have delivered 8.2 billion

cumulative treatments to people living in endemic areas [2], whilst fewer lymphedema and hydrocele patients have accessed MMDP services in the same period.

According to the World Health Organization (WHO), MMDP services should be included in the basic primary healthcare package [4–7]. However, progress towards establishing and streamlining MMDP services still remains considerably slow, particularly in the African region, which accounts for a considerable proportion of the LF burden [2]. This is evident in the number of countries that have put in place reporting mechanisms for patients with lymphedema and hydrocele. For instance out of 34 LF endemic countries in the WHO Africa region, only 22 are reporting on lymphedema patients and 23 on hydrocele patients [2]. Global efforts to eliminate LF through MDA are likely to scale down beyond 2030, and strategic direction of diseases control efforts will most likely pivot towards the provision of MMDP services.

In Zambia, LF is a public health concern, as 87 of 118 districts are considered endemic with the prevalence of the circulating filarial antigen above 1.5% [2,8]. MDA for LF was first piloted in Western Province in 2014 and then scaled up nationally in 2015 with annual rounds running until 2018. In 2018, national coverage for MDA was reported at 90.8% [2]. Morbidity mapping of LF patients has been conducted concurrently with the MDA rounds. Results from the LF mapping exercise have shown that there are many cases of hydrocele and lymphedema spread across all ten provinces in the country [8]. However, there is limited information of what MMDP services are available as well as where and how LF patients are accessing them. Furthermore, LF patients' health beliefs and health seeking behavior towards MMDP services remains largely undocumented in Zambia.

The Zambia Elimination of Neglected Tropical Diseases National Masterplan (2019–2023) places huge emphasis on delivery of MMDP services in endemic districts. However, Zambia currently lacks a comprehensive national MMDP strategy for LF or suitable indicators to monitor progress in service provision [8]. The Ministry of Health (MoH) has been working in collaboration with the University of Zambia, School of Public Health as part of efforts to generate evidence to guide the formulation of the national MMDP strategy. They jointly piloted a program to identify mechanisms through which MMDP services can be streamlined and integrated into primary health care systems. This study presents the findings of a preliminary baseline assessment conducted prior to the development and implementation of the pilot program. Specifically, this paper reports on the health beliefs and health seeking behavior towards MMDP services for LF among community members and health providers in Luangwa District, Zambia.

## Methods

### Ethics statement

Ethical approval was sought from University of Zambia Biomedical Research Ethics Committee (REF.017-11-18) and the National Health Research Authority under the Ministry of Health, Zambia. All participants were informed of the purpose of the assessment, details of study procedures including freedom to withdraw, potential benefits and risks, prior to the commencement of data collection. Thereafter written informed consent was obtained as all participants were 18 years and above.

### Study design

An exploratory qualitative research was undertaken as part of a larger baseline formative assessment to inform the development and implementation of an integrated health systems intervention to improve access to MMDP services for LF patients in Luangwa District. Due to paucity of information within the Zambian context on MMDP, this design was felt to be most

appropriate to fully explore community perspectives and common management practices of the LF chronic manifestations in the district. The study was conducted between February and April 2019.

## Study setting

**Health systems context.** Local health systems face numerous challenges in the delivery of MMDP services. Very few rural health centers prioritize the provision of the basic MMDP package of services. Available primary health services are general in nature without specific stand-alone activities for chronic manifestations of LF such as hydrocele and lymphedema. A limited number of health facilities have adequate resources to conduct hydrocelectomies as well as provide lymphedema management services. In the case of lymphedema, the most readily available services are pain relief and general health education with IEC materials on display in busy outpatient areas. Some health facilities also provide antibiotics to prevent secondary infections as a result of acute attacks as well as lymphatic draining to reduce fluid density. Furthermore, there are existing referral systems is in place. Hydrocele cases are referred to the two second level hospitals in the district; Katondwe Mission and Luangwa Boma, which are equipped to perform hydrocelectomies. Severe cases are referred to the University Teaching Hospital in Lusaka District. Despite surgical interventions being available at health facilities, uptake has been reported to be very low. In addition, local healthcare providers and community health workers (Community Health Workers) responsible for conducting case identification and management have limited training on provision of MMDP services.

## Participant recruitment

In order to select the most appropriate health facility catchment areas from which data would be collected, the study team conducted a morbidity mapping exercise as available records from the Ministry of Health were not up to date. This mapping exercise consisted a census of patients with chronic manifestations of LF conducted by Community Health Workers who are usually engaged by the health facilities in drug distribution during MDA for LF campaigns. The Community Health Workers received training on LF case identification and went from household to household recording all patients who exhibited signs of elephantiasis, lymphedema and hydrocele. There was a total of 237 cases identified during the mapping exercise in the district; 27 lymphedema, 199 hydrocele and 7 with both hydrocele and lymphedema. Based on the census, eight health facility catchment areas which had the highest number of LF patients were selected as study sites. These were Luangwa Boma, Mpukha, Katondwe, Kanemela, Chitope, Kasinsa, Mandombe clinics and Luangwa District Hospital. The different participant category numbers across the study sites were recruited based on the guidance from the district health office, facility in charges and Community Health Workers. Whilst this study focuses on the formative exploratory qualitative research, the results of the baseline census of LF patients are reported elsewhere.

## Data collection

Data was collected through a series of focus group discussions (FGDs) and in-depth interviews (IDIs). The number of participants for both FGDs and IDs was determined by the principle of saturation of in information in qualitative research. Interview and FGD guides contained questions on the causes and manifestations of the disease, cultural beliefs, disease management practices, health seeking behavior as well as factors that affect how patients access to healthcare [7] (Finalized tools provided as S1 Data collection tools). The interviews were conducted by trained research assistants from the University of Zambia School of Public Health under the

supervision of PM, JMZ and GS. Prior to data collection, the guides/tools were piloted to ensure their suitability by adapting questions where necessary. All FGDs and IDIs were audio recorded with the consent of the study participants. In addition, field notes were taken during the course of the interviews. Data collection was done in English and Nyanja to ensure that all participants were able to articulate their perspectives as comprehensively as possible. The recordings were transcribed verbatim and those done in Nyanja were translated to English.

**Focus group discussions.**   There were 4 mixed focus group discussions (2 male and 2 female) held with community members and LF patients in two health facility catchment areas; Luangwa Boma and Mphuka, which had reported the highest number of cases in the district after the morbidity mapping exercise. The participants were purposively sampled from the communities around the health facilities' catchment areas. Recruitment was done by the Community Health Workers and health care providers attached to the two health facilities. Participants were invited to take part in the study via telephone by the health facility in-charges. On average each FGD comprised nine participants aged between 18–50. The FGDs were differentiated by gender due to prevailing cultural beliefs in the district surrounding the ease with which community members could talk about hydrocele which is considered a sensitive topic of discussion. Such separation encouraged community members to freely express themselves. The FGDs were conducted at the health facility, but away from patient areas to ensure the privacy of the participants and avoid interruptions. The FGDs lasted between 1 hour 30 minutes to 2 hours.

**In-depth interviews.**   A total of twenty-six interviews were conducted with district neglected tropical disease focal point persons (n = 2), community health workers (n = 8), health facility staff (n = 8) and traditional leaders (n = 8). The participants for the IDIs were purposively sampled based on their involvement in the implementation of LF elimination programs, and were drawn from across the 8 health facility catchment areas that had the highest number of LF patients. The interviews with the community and traditional leaders were centered around exploring community perspectives of LF, stigma, social support structures, health beliefs, health seeking behavior and accessibility of MMDP services in their respective communities. The interviews with the health providers focused on their training, knowledge of, and factors shaping their ability to effectively provide MMDP services. Participants were invited to take part either through telephone or face to face interviews. Forty-five minutes to one-hour IDIs were conducted at the selected health facilities and the Luangwa district health office.

## Data analysis

A thematic analysis approach, which is a technique of exploring relationships and patterns in the qualitative data was used [9]. Reports from the field notes and preliminary reading of the transcripts informed the development of the initial coding tree that was then imported to NVivo 12 software (Table 1). Subsequent coding and iterative discussions among the authorship team allowed for additional modifications of the coding structure. We used both inductive and deductive coding approaches to ensure that both broader level and emergent sub-themes in the data were exhaustively captured. Code reports on each of the thematic areas were then analyzed to understand patterns and relationships in the data.

After completion of the data analysis, a stakeholder meeting was held to validate the findings of the study and to confirm that the information captured from the participants was accurately reported. A total of fifty participants took part in the validation meeting including Local Chiefs, traditional leaders, Ministry of Health staff and Community Health Workers. The validation meeting provided an opportunity for further interrogation and alignment of the study findings. More so, it provided a platform to clarify participant perspectives on some of the findings.

Table 1. Summary of qualitative coding tree.

| Broader theme | Sub-theme | Sub-theme |
|---|---|---|
| Health beliefs | Local terminologies | Tumbu |
| | | Nchofu or musakasa |
| | Cause of LF | Undercooked food |
| | | Herbal aphrodisiacs |
| | | Hereditary |
| | | Witchcraft |
| | Signs and symptoms of LF | |
| Health seeking behavior | Traditional healers | First care option |
| | | Effective treatment |
| | Fear of hydrocele surgery | Lead to infertility |
| | | Permanent disability |
| | | Removal of testicles |
| | Home based care strategies | |
| Barriers to seeking care | Health system barriers | Distance to the health facility |
| | | Lack of awareness of MMDP services |
| | | Cost of accessing care |
| | Cultural barriers | Negative gender norms |
| | | Negative social norms |
| | | Stigmatization of LF patients |

## Results

The study results are presented according to the views of various categories of participants; community members, LF patients, healthcare providers, Community Health Workers and traditional leaders. Verbatim quotes are also presented to provide context on the participant perspectives. Most of the views on key thematic areas were similar among the different category of participants.

### Health beliefs of lymphedema, hydrocele and elephantiasis

During the FGDs, participants pointed out the local Nyanja names associated with the symptoms of the different manifestations of LF; *tumbu* or *nchofu* for hydrocele and *musakasa* for all forms lymphedema. The signs and symptoms of the different manifestations of LF chronic conditions were well known by the community members who were able to describe them during the FGDs.

> "This disease of lymphedema, I used to see people swelling the testicles, legs and to my side at first, I took it to be normal like that's how they were born. All the years I took it like that." (FGD1, Community member)

The most common mentioned causes of LF included eating food that had not been warmed properly, contact with animal faeces, using traditional medicine such as local herbal aphrodisiacs (*mutoto*), and sexual contact with a woman who was menstruating or who had a miscarriage. Other perceived causes included witchcraft, men using pounding sticks after they have been used by women, sitting on a chair recently used by an LF patient, sitting on a stone that women use during food preparation, and children vomiting on their mothers during breastfeeding. In addition, there was a belief that LF chronic manifestations were hereditary. Only a

handful of the FGD participants correctly identified mosquitoes as the disease-causing vectors, and that LF could be prevented by taking medication.

> "Sleeping with a lady who was pregnant and whose baby dies and they are found in clubs or bars without you knowing anything, you approach her and when she accepts to have sex with her, you end up getting the disease from her." (FGD2, Community member)

> "I didn't know anything I could only see a woman her breast gets swollen, it could be a leg and hand. I thought it was a different disease. I didn't know that the mosquito brings the disease." (FGD1, Community member)

Community actors such as traditional leaders reiterated some of the beliefs that came up during the FGDs. For instance, one of the chiefs stated that;

> "……. Here we believe that if you are passing where there are feces of animals especially this rain season whether bush animals or village animals and it can be goats, cattle, sheep, pigs, elephants or other animals the disease goes inside the nails making your legs swollen so even if you treat hydrocele it can't be healed and that person will die at that old age with swollen legs like that…." [IDI5, Traditional Leader].

Nevertheless, knowledge of the causes of LF among community actors such as traditional leaders, community-based volunteers and healthcare providers differed greatly depending on their level of involvement in LF disease control activities such as MDA. Most of them acknowledged that they had limited knowledge of how to manage the disease. Healthcare providers who came into regular contact with patients coming to the facility or through community case identification exercises were better at identifying signs and symptoms of lymphedema, elephantiasis and hydrocele. One health provider indicated;

> "….Even the skin changes and it looks like it doesn't not look to be a normal skin. It changes the color and it becomes hard. For lymphedema even the limbs, arms legs differ to a normal one. They become different in size….[IDI2, Health provider]."

## Healthcare seeking behavior

It was apparent from the FGDs and interviews that, patients' decision to seek care from both traditional healers and from the health facility was linked to the prevailing ideas about the causes of the disease. Due to the communal belief that LF is hereditary, patients who had seen their family members exhibiting symptoms and not seeking appropriate care did not see the need to go to a health facility. Moreover, community members who viewed it as a disease that arose due to witchcraft rather than mosquitoes were more likely to seek help from traditional healers. There was also a common belief that the remedies provided by the traditional healers were more effective and permanent solutions in comparison to what was provided at the health facilities. One of the participants who had lymphedema pointed out;

> "I have had three years with this disease and here at this clinic they just give me Panadol when I come so I stopped coming because Panadol doesn't work." (FGD 2, Community member)

The study participants indicated that some lymphedema patients would first go to a traditional healer and only visit a health facility once their symptoms became more severe. This was

because the tattoos and herbs administered by the traditional healers would exacerbate the acute attacks due to LF forcing the patients to go to the health facility for specialized treatment.

> "With traditional beliefs you will think let me go and look for traditional medicine and put traditional tattoos. Now when they put tattoos instead of the legs healing they start swelling because traditional tattoos now start bringing sores because people are different, some of them it just starts swelling without any pain and then you go to see some with doctors or to the clinic." [IDI1, Community Health Worker].

Very few of the community members were able to identify any homebased care strategies that could prevent the progression of lymphedema and reduce the occurrence of acute attacks. With regards to hydrocele, it was rare for patients to go to health facility to seek interventions such as aspiration of fluid or hydrocelectomies because they were afraid of undergoing the surgical procedure, being rendered infertile or incapacitated. Some participants narrated that they thought the surgical procedure was actually meant to remove the testicles.

> "Some people are scared to go for operation to say they will be operated, they think when they do they can be gone for good and secondly they say when you are operated you will not have children anymore as a result people go for traditional medicine." (FGD3, Community member)

The patients reported that they were afraid to go to the health facility because they felt that their conditions had become so advanced that any treatment they received would not lead to an improvement of their symptoms. Healthcare providers and Community Health Workers pointed out that as a result of this, it was common, for hydrocele patients to come to the health facility with very huge swellings. The loss of hope hampered patient motivation to maintain home based care practices which are critical to ensuring that lymphedema does not progress to elephantiasis, and that surgical interventions are undertaken early for hydrocele.

> "Say even if am to go to the clinic I won't be healed am already disabled or paralyzed so even if am to go to the clinic I won't be helped, others regardless of the condition being severe, they would remain home and say am disabled already." [IDI3, Community Health Worker].

## Health system and cultural barriers to seeking healthcare

**Distance to health facility.** Luangwa District is very remote and a portion of the district is covered by the Luangwa National Park. As a result, some communities in the district have to travel long distances of up to 20 kilometers and more to the nearest health facility and their access may be inhibited by wildlife attacks such as elephants from the Luangwa National Park. In addition, roads to the health facilities are sometimes impassable and the most common means of transport is bicycles which are inappropriate to transport lymphedema and hydrocele patients. As such some patients choose to stay home rather than go to the health facility. The district also borders Mozambique and Zimbabwe and patients who may be involved in economic activities which require them to travel across the borders are often missed out when healthcare providers conduct outreach and follow up visits at community level to provide MMDP services as they may not be found at their homes.

> "Some villages are quite on remote areas. Hence depending on the transportation, they may not be able to come from here and also some they come from across our neighboring country across the river in Mozambique." [IDI3, Healthcare Provider].

**Lack of awareness of existing MMDP services.** A recurring theme across the FGDs and interviews was that most community members were not aware that there are MMDP services available at the health facilities or that there are home based care strategies that lymphedema patients can use to prevent the worsening of their conditions. They indicated that the information that was provided to them by the Community Health Workers was mostly focused on the importance of taking part in annual drug distribution exercises during MDA for LF campaigns. Furthermore, other ongoing community sensitization exercises tended to focus on diseases such as malaria, HIV/AIDS and maternal, newborn and child related conditions, as LF is not perceived to be a public health priority. Nevertheless, the study participants reported that the healthcare providers and Community Health Workers had now begun conducting community outreach and health talks with the help of LF patients who would act as champions to encourage more community members to utilize MMDP services.

> "What makes these people to come to the clinic, maybe a fellow patient went to the clinic and was assisted then the information is spread to fellows. Then we encourage them that you find you have the right to share with friends that I was helped in this way so that those who are shy can be motivated and go to the clinic openly and express their problems." [IDI3, Community Health Worker].

**Costs of accessing healthcare services.** Majority of the residents in the district are poor and rely on subsistence farming and fishing for sustenance. As such, when deciding on how best to prioritize available financial resources, they opted to spend on food and basic necessities before considering health care, particularly for conditions that they thought were incurable. For this reason, some patients feared going to the health facility to access MMDP services due to the perceived costs they would incur to not only access, but also getting to the health facility considering their disability.

> "Yes, some they think that way. That they will pay for the operation, they think that if doctors refer me to the theatre where am I going to get the money so it's better I just stay with my swollen legs." [IDI5, Traditional Leader].

There was a perception among community members that they were expected to pay for treatment services such as hydrocelectomies which dissuaded them from seeking care. Furthermore, the costs associated with moving from their houses to the health facilities in their conditions of health hampered access to care. For example, in the event that a patient was found to have a severe case of hydrocele or lymphedema requiring specialized treatment at one of the referral facilities, the associated costs such as hiring transportation and out of pocket hospital expenses acted as barriers to accessing care. Families that had LF patients were sometimes forced to save money over long periods of time to enable them access care.

> "So, for them they think, if I start going there with my cost of living, it is difficult to find money, so when they think I start going to the hospital and maybe there are also some payments at the hospital, no its better I just don't go." [IDI1, Community Health Worker].

## Gender and social norms

According to the healthcare providers, male hydrocele patients were less likely to visit the facility and speak freely about their condition. The participants indicated that one of the most common prevailing cultural belief was that conditions to do with genitalia should not be discussed

openly or even shown to members of the opposite sex. As such hydrocele patients were embarrassed to access services and even when they did, they did not easily open up, especially if they were being attended to by female healthcare providers.

> "**P**eople are not open to explain the problem they have especially if a woman is interviewing a man it becomes a big challenge. Others are known that he has such a problem but talking to him, he refuses completely." [IDI2, Community Health Worker].

Nonetheless, having hydrocele was viewed as a marker of high social standing, and men who have it were more likely to be chosen as headmen as they are perceived to be old and wise. This also acted as a deterrent to patients seeking care.

> ". . .. When they get sick they don't even go to the hospital because they are respected so much. Yes, they respect him a lot even if you reach somewhere he will be the first one to be given the chair just because of what they are seeing, respecting him a lot that one who doesn't have, so mostly if you become a headman they believe you are supposed to have hydrocele and you should become the head of the family you are supposed to have that so that even if you this is our leader. . ..."[KII2, Traditional Leader].

## Fear of stigmatization

Fear of stigmatization was reported to inhibit LF patients' ability to seek care. During regular case identification exercises by the healthcare providers, known hydrocele patients, who were approached for referral to the health facility would either deny having the condition, or request to be talked to in private because they were afraid of being laughed at by other community members.

> "Especially if he has hydrocele they start laughing at him so you will find sometimes he can't even be open and be free to go to the hospital because of the fear that people will laugh at me." [KII2, Traditional Leader].

> "They tell you we can talk in a hidden place or maybe you just come later, because there are people present." [IDI1, Community Health Worker].

## Discussion

Some of the perceived causes of hydrocele and lymphedema included witchcraft, direct contact with infected individuals or objects as well as it being heritable, which matches results from similar studies [10–12]. This poor understanding of the connection between mosquitoes as the causes of filarial infection and the advanced stages of the disease has also been found elsewhere [10,13,14]. Further, knowledge of morbidity management practices and services was shown to be very minimal, which calls for increased demand generation efforts as these services help to ease the burden of disability and improve the quality of life [15]. Efforts should therefore be channeled towards creating more community awareness of existing MMDP services, as well as providing information that dispels some of the misunderstandings around LF causes.

Health education campaigns through the existing network of Community Health Workers have been found effective and should be promoted [16]. Community Health Workers, often the main source of information have been instrumental in creating demand as well as influencing behavior change in most community health programs [13,16]. However, the current

situation in Zambia is such that most of the information that the Community Health Workers provide is limited to creating awareness about MDA for LF, leaving out critical but important information on how to manage and reduce disability due to LF. The Zambia national MMDP strategy should seek to promote integration of information on MMDP services in daily Community Health Workers programing to ensure that patients with chronic manifestations of LF are not left out. Further, the MMDP strategy must ensure that a training curriculum that equips Community Health Workers with information on key MMDP services is developed and availed across the health system.

Traditional healers still remain the preferred first line of care for most LF patients living in the community as reported in other studies [10,13,17,18]. Most patients only go to the health facility once the symptoms have progressed and the pain worsened. Consequently, this has a negative bearing on their treatment outcomes. The Zambia national MMDP strategy should prioritize the engagement of traditional healers given their influence on health seeking behavior amongst the LF patients. Their inclusion in the delivery of MMDP services is of critical importance as they can act as champions who may refer patients to the facility for more effective treatment and management of the LF chronic manifestations.

Motivation to seek care amongst most of the patients with LF chronic manifestation often wanes once they realize that their conditions can only be managed over time and are incurable [11,13,18,19]. There is a tendency to question the value of seeking care knowing very well that they will not be cured. This has been reported in other studies, particularly with regards to MDA for LF chemoprevention, where those who already have the disease abstain from taking the drugs. Demotivation is also manifest in the poor uptake of hydrocelectomies, where patients are not only afraid of the surgery itself, but also, that they may become sterile. The Zambia national MMDP strategy should empower and motivate, patients, households and communities to seek these readily available free-of-cost primary healthcare services.

Health system and cultural barriers to seeking MMDP services including long distances to the health facilities, lack of awareness, perceived costs of services, and fear of stigmatization have also been reported in other studies [13,17,20,21]. However, addressing some of these barriers will require system wide policy level interventions that recognize LF as more than a health sector problem. Efforts to lighten the burden of LF in our communities must go beyond just providing MMDP services at the facility, but also promote community empowerment, possible access to welfare and economic opportunities for those suffering from LF chronic manifestations [22]. Health systems strengthening efforts must guarantee capacity to provide and sustain MMDP services to communities in need through providing appropriate leadership and training. For example, capacity for provision of psychosocial counseling and rehabilitation services to LF patients must be enhanced through continuous training and supply of required commodities. Furthermore, enhanced high level political commitment, wider publicity, effective referral systems, integration of MMDP with other disease management services and collaboration with research organizations are essential in strengthening local health systems capacity to provide MMDP services [23].

MMDP services remain a critical component of the global strategy to address the lymphatic filariasis burden in most low-income settings. Whilst most efforts have predominantly been centered around MDA for LF, MMDP services provide a platform to complement these efforts. Studies have documented that creating demand for MMDP services has also had a positive impact on community acceptability of MDA for LF [24]. Morbidity management programs for lymphedema and hydrocele have been reported to increase community support for and hence participation in MDA for LF [25]. These programs provide training on self-management of lymphedema for patients and hydrocele surgical operation for the healthcare providers. Community knowledge of available care, including surgery for hydrocele patients

motivated people to participate in MDA for LF. Lymphedema management programs also provided patients with a platform to share information with other community members about the disease and the benefits of the drugs [26].

## Strengths and limitations of the study

This was an exploratory qualitative study that sought to gather data on one of a most neglected, but prevalent public health problem in Zambia. Data were gathered from varied categories of participants/actors from both the health systems, community and household levels to provide information on some of the underlying issues and multitude of factors affecting access to MMDP services in Zambia. However, one of the major limitations in the study was that only a small fraction of LF patients was interviewed as most of their data was captured using a close ended survey that is reported elsewhere. But even with this limitation, the patients as well as other categories of participants were able to provide the required information. Another limitation was that the sample of Luangwa District may not represent views of other LF endemic areas in Zambia, but still provides a critical learning points in efforts to create national strategy for MMDP services.

## Conclusion

This study found that hydrocele and lymphedema are well known among communities living in Luangwa District, with their health seeking behavior towards MMDP services largely driven by the perceived causes of LF. Ongoing community sensitization as well as ensuring that healthcare providers and Community Health Workers working in endemic districts are equipped with the necessary tools to deliver MMDP services is required. Health education campaigns at community level should address patients concerns surrounding access to care including reduced motivation to maintain lymphedema management practices and fear of taking up hydrocele surgery due to perceived side-effects of the procedure. In addition, health education campaigns should seek to create awareness of existing free-of-charge MMDP services. There is a need to strengthen referral systems to ensure LF patients not only get appropriate care but are also regularly followed-up and linked to other critical social services to enhance their quality of life. Furthermore, MMDP programs should embrace strategies that seek to empower and motivate LF patients and their families to overcome some of the health systems and cultural barriers to accessing care.

## Supporting information

**S1 Data collection tools. Interview and focus group discussion guides.**
(DOCX)

## Acknowledgments

We would like to acknowledge Edson Musonda and the entire Luangwa District Health Office as well as Frank Shamilimo- the Lymphatic Filariasis focal point person at the Ministry of Health, Zambia who facilitated the study process.

## Author Contributions

**Conceptualization:** Patricia Maritim, Adam Silumbwe, Joseph Mumba Zulu, George Sichone, Charles Michelo.

**Data curation:** Patricia Maritim, Adam Silumbwe.

**Formal analysis:** Patricia Maritim, Adam Silumbwe, Joseph Mumba Zulu, George Sichone, Charles Michelo.

**Funding acquisition:** Patricia Maritim, Adam Silumbwe, Joseph Mumba Zulu, George Sichone, Charles Michelo.

**Investigation:** Patricia Maritim, Adam Silumbwe, Joseph Mumba Zulu, George Sichone, Charles Michelo.

**Methodology:** Patricia Maritim, Adam Silumbwe, Joseph Mumba Zulu, Charles Michelo.

**Project administration:** George Sichone, Charles Michelo.

**Resources:** Charles Michelo.

**Supervision:** Joseph Mumba Zulu, Charles Michelo.

**Validation:** Patricia Maritim, Joseph Mumba Zulu, Charles Michelo.

**Visualization:** Patricia Maritim.

**Writing – original draft:** Patricia Maritim, Adam Silumbwe, Joseph Mumba Zulu, George Sichone, Charles Michelo.

**Writing – review & editing:** Patricia Maritim, Adam Silumbwe, Joseph Mumba Zulu, George Sichone, Charles Michelo.

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
