## [Decision Letter · Decision Letter 0]

17 Apr 2020

Dear Ms Maritim,

Thank you very much for submitting your manuscript "Access to morbidity management and disability prevention services for lymphatic filariasis in Luangwa district, Zambia: A mixed methods study." for consideration at PLOS Neglected Tropical Diseases. As with all papers reviewed by the journal, your manuscript was reviewed by members of the editorial board and by several independent reviewers. In light of the reviews (below this email), we would like to invite the resubmission of a significantly-revised version that takes into account the reviewers' comments. 

We cannot make any decision about publication until we have seen the revised manuscript and your response to the reviewers' comments. Your revised manuscript is also likely to be sent to reviewers for further evaluation.

Sincerely,

Samuel Wanji

Guest Editor

Francesca Tamarozzi

Deputy Editor

Reviewer's Responses to Questions

**Key Review Criteria Required for Acceptance?**

**Methods**

-Are the objectives of the study clearly articulated with a clear testable hypothesis stated?

-Is the study design appropriate to address the stated objectives?

-Is the population clearly described and appropriate for the hypothesis being tested?

-Is the sample size sufficient to ensure adequate power to address the hypothesis being tested?

-Were correct statistical analysis used to support conclusions?

-Are there concerns about ethical or regulatory requirements being met?

Reviewer #1: objectives are clearly stated

The study design is questionable 

The study population is not clearly described for certain sections of the methodology (focus group discussions)

Sample size for patients with lymphoedema is low and not adequate

Reviewer #2: The study has clearly articulated objectives and subsequently addressed by the study. The study design is appropriate to address the objectives. The study setting should be described well. They have used all cases in the cross-sectional study, but it was not clear how they decided to have 4 FGDs and 20 in-depth interviews.

Reviewer #3: Overall the methods are satisfactory. However, the questions of the survey need to be included (with in text or the survey and all questions as a supplementary file) so it is clear what the results are presenting. The 

results need to be more systematically presented perhaps with a could of sub-headings. 

The population is defined for one survey, but how were the people and places for the FGD and interveiws selected and how were the number of FGDs and interviews derived? Maybe a reference for FGDs are required on line 154. How were the participants recruited from the community? Please add in a little more detail.

**Results**

-Does the analysis presented match the analysis plan?

-Are the results clearly and completely presented?

-Are the figures (Tables, Images) of sufficient quality for clarity?

Reviewer #1: Results were erroneous incomplete and not clearly presented. Require additional Tables with quantitative data

Reviewer #2: Yes, the analysis plan clearly matches the results.

Reviewer #3: Overall the results need to be presented in a more succinct and structured way and better linked with the methods (e.g. similar sub-heading would be helpful). 

A table summarising key results of the community survey would help to reduce text and give more meaning the proportions. 

Need to make it clearer the information coming from community FGDs or health workers in each section.

Line 232 – a break down of each condition by gender should be included (e.g. how many lymphoedema cases were male/female). Include mean age of men and women

There are no figures presented - but I wonder if the main themes and subthemes could somehow be presented as a summary organagram/venn diagramm or piechart/circle or similar to highlight the range of issues - it iwl be easier for the reader to glance at a figure and see this

**Conclusions**

-Are the conclusions supported by the data presented?

-Are the limitations of analysis clearly described?

-Do the authors discuss how these data can be helpful to advance our understanding of the topic under study?

-Is public health relevance addressed?

Reviewer #1: The conclusions are not based on the data presented. No reference to study limitations. The data presented give some idea of the factors that are affecting the utilization of MMPD services .

Reviewer #2: While the conclusion are supported by the presented data, the authors did not indicate the limitations of the study and its implication in the interpretation of the findings. The implication of the current study in advancing our understanding of the topic is not discussed well. The public health relevance and implication for programme implementation and clinical practice is not discussed.

Reviewer #3: (No Response)

**Editorial and Data Presentation Modifications?**

Reviewer #1: (No Response)

Reviewer #2: (No Response)

Reviewer #3: (No Response)

**Summary and General Comments**

Reviewer #1: Reviewer

The authors are presenting the findings of a KAPP (knowledge attitude, practices and perceptions) study on lymphatic filarial disease aetiology and management among three stakeholder groups ( patients, community and health providers) in an endemic setting in the district of Luanga in Zambia where the majority appear to be of low educational attainment with more or less primitive health care facilities. The writing needs to be improved overall so as to improve the clarity of the message conveyed, Results documented were insufficient and erroneous and not presented clearly. 

1. I feel that the title “ Access to morbidity management and disability prevention services for lymphatic filariasis in Luangwa district, Zambia: A mixed methods study” is somewhat inappropriate as the manuscript does not detail the availability of MMDP health services in the region (number of health facilities that can perform hydrocelectomies, facilities which provide limb care etc ). Without knowing the availability of baseline facilities I feel that you cannot discuss access (defined as “opportunity to reach and obtain appropriate health care services in situations of perceived need for care” ) to these services. What the authors are describing are the health seeking behavior of patients and factors that influence the behavior patterns. Overall writing is poor and the meaning of certain statements are rather ambiguous and not clear. This is just one of the instances, Eg, “…..of the MMDP intervention such as the design training curriculums, health education and….” .There are many more grammatical errors which need to be attended.

2. The KAPPP findings were interesting but somewhat vague due to lack of quantitative data. I feel that the Results section need to be re-written as presentation of research findings is not very clear and rather confusing as outcomes of all surveys are presented together and discussed in this section. Furthermore the data given in the results section appear to be incorrect. Percentages without the actual values of the variable are unacceptable as there is no way to verify the accuracy of data. Perhaps using tables to present the quantitative data may increase the clarity of the findings. The results of the 3 surveys; patient KAPPP, FGD and interviews ( specify the group,.. primary health care providers/ Key informants/ community leaders?) if presented separately may be more clearer to the reader as well as the authors. 

Is it possible to specify the key areas that were investigated by the KAPP survey, the questions included to the questionnaire ? used on the patients, In the FGDs were the same questions asked from the community was it approached differently? Was the approach similar for all FGDs? With regard to interviews detail the core areas that were covered. 

3. The basic results documented are erroneous and contradictory,

 Lines 229 & 230; Patient characteristics There were 237 pts…….199 hydrcoele, 27 lymphoedema, 7 both lymphoedema and hydrocele, addition brings the total to 233, 

Line 320 says of the 22 patients with lymphedema? I think even the authors are not clear about how many patients had lymphoedema. The number of lymphoedema patients are rather low to discuss the morbidity management measures practiced by them. Presenting the results as percentages is not acceptable, include the actual value of the variables as well. 

4. Line 239, Knowledge of lymphedema, elephantiasis and hydrocele,

It is important to document the disease knowledge among patients rather than community, there is no data on this aspect.

With regard to FGD, what was the composition of the sample? (It says a convenient sample) how many patients were included? What was the age range?

5. Line 354, Social norms. Appear to be contradictory” Having hydrocele is viewed as a marker of high social standing and men who had it are more likely to be chosen as headmen as they are perceived to be old and wise” and the statement “Though LF patients were considered to be in a pitiful state” and lines 376-380 under Stigmatisation ” Fear of stigmatisation also inhibits patients’ ability to seek care….would either deny having the condition or request to talk to them in private because they are afraid that they would be laughed at if other community members found out that they had hydrocele” Are these statement derived from the FGD or are these the authors views?

6. . Beliefs (line 342) were these derived from FGDs or Patient surveys or past publications? (Shawa et al 2013)

7. The manuscript is too lengthy and include a lot of detail on implementation research approaches ( lines 102-126 ) patient access to care frame work guidelines (191-195 )but detail (data) on the current study outcomes and the Discussion are inadequate. The authors should compare and discuss the similarities and differences of their results with past reports (publications) on LF in Zimbabwe and elsewhere, discuss the limitations of the study etc.

Reviewer #2: This is an important manuscript on Access to morbidity management and disability prevention services for lymphatic filariasis in Luangwa district, Zambia: A mixed methods study. The authors used both quantitative and qualitative methods to assess the barriers and opportunities for integration of the MMDP services. The authors conducted a cross-sectional survey of 237 patients, 4 Focus group discussions and 20 interviews. This a comprehensive study with many important finings for planning MMDP programme and educational materials. Having said that I have the following specific comments to improve the work further. 

1. The manuscript is too long given the number of focus groups and in-depth interview conducted. I would suggest the authors to focus on the key findings than listing everything here. 

2. In the study setting please give clear description of the study setting. When was the LF treatment started in the district? How many health facilities are there in the district? What type of health facilities? How is the function of the health system organized in the district etc. 

3. The discussion part is very brief I would suggest the authors would include, comparing their findings with previous studies, the key limitations of their study. The implication of the current study to programme planning and implementation and some recommendations.

Reviewer #3: Overall a reasonably written paper addressing a neglected topic, especially in Zambia. The use of mixed methods is good, however as little more delination between the different groups woudl be helepful so it is easier to see who is saying what. 

In the abstract the findings could include more hard data/figures as it is a little vague and the challenges should link to the paper key themes so there is consistency

The discussion needs more references supporting the statements. 

MMDP - inconsistency in how it is presented i.e. sometime abbreviated but not always. Please check all 

The authors may want to highlight the economic benefits of surgery as published in PloS recently https://www.ncbi.nlm.nih.gov/pubmed/32210436 . This paper also has a number of references that the authors here could use/include to better back some of the statements.

In general there is a lack of references to support the paper

PLOS authors have the option to publish the peer review history of their article (what does this mean?). If published, this will include your full peer review and any attached files.

Reviewer #1: Yes: T. G.A. Nilmini Chandrasena

Reviewer #2: Yes: Kebede Deribe

Reviewer #3: No
---

## [Decision Letter · Decision Letter 1]

4 Sep 2020

Dear Ms Maritim,

Thank you very much for submitting your manuscript "Community perspectives towards morbidity management and disability prevention for lymphatic filariasis in Luangwa district, Zambia: A qualitative study." for consideration at PLOS Neglected Tropical Diseases. As with all papers reviewed by the journal, your manuscript was reviewed by members of the editorial board and by several independent reviewers. The reviewers appreciated the attention to an important topic. Based on the reviews, we are likely to accept this manuscript for publication, providing that you modify the manuscript according to the review recommendations. 

Sincerely,

Francesca Tamarozzi

Deputy Editor

Francesca Tamarozzi

Deputy Editor

Reviewer's Responses to Questions

**Key Review Criteria Required for Acceptance?**

**Methods**

-Are the objectives of the study clearly articulated with a clear testable hypothesis stated?

-Is the study design appropriate to address the stated objectives?

-Is the population clearly described and appropriate for the hypothesis being tested?

-Is the sample size sufficient to ensure adequate power to address the hypothesis being tested?

-Were correct statistical analysis used to support conclusions?

-Are there concerns about ethical or regulatory requirements being met?

Reviewer #1: The objectives are rather vague.

A qualitative study was done which may not be the best way to address the objectives stated. Perhaps as a preliminary fact finding mission it may be acceptable.

The population is described and is appropriate but the sample size of LF disease patients (lymphoedema an important category of MMDP stakeholder ) is grossly inadequate.

Since it is a qualitative study there are no numerical values and no statistics in data analysis. Terms used to describe study outcomes in the Results section are very vague," common"," few" etc which is direct effect of the study design.

No ethical concerns

Reviewer #2: The objectives of the study are clearly stipulated in the manuscript. The authors used appropriate design to address the objectives of the study. This has also been aligned with the title of the study. The authors conducted FGD and in-depth interviews to collect data. It would be better to mention how they determined the number of FGDs and in-depth interviews. Was it based on saturation of the data? The study is qualitative and the authors have used appropriate software to analyse the data. The authors have ethical clearance from Ethical approval was sought from University of Zambia Biomedical Research Ethics Committee (REF.017-11-18) and the National Health Research Authority under the Ministry of Health, Zambia.

Reviewer #3: (No Response)

**Results**

-Does the analysis presented match the analysis plan?

-Are the results clearly and completely presented?

-Are the figures (Tables, Images) of sufficient quality for clarity?

Reviewer #1: Since it is a qualitative study there are no numerical values and no statistics in data analysis. Terms used to describe study outcomes in the Results section are very vague," common"," few" etc which is direct effect of the study design

Table had some concerns which were included to reviewers comments

Reviewer #2: Yes

Reviewer #3: (No Response)

**Conclusions**

-Are the conclusions supported by the data presented?

-Are the limitations of analysis clearly described?

-Do the authors discuss how these data can be helpful to advance our understanding of the topic under study?

-Is public health relevance addressed?

Reviewer #1: Conclusions are inadequate

The limitations of the study design need to be discussed.

Public health relevance is discussed but there is room for expansion

Reviewer #2: The conclusions made are supported by the data presented in the manuscript. The authors have clearly discussed the strengths and limitations of the study. They have also highlighted the implication of the study. Nonetheless, strengthening the discussion of the public health relevance of the study would be important.

Reviewer #3: (No Response)

**Editorial and Data Presentation Modifications?**

Reviewer #1: minor modifications suggested. But not entirely convinced that it is worthy of PLOS NTD journal acceptance

Reviewer’s responses to the edited version titled “Community perspectives towards morbidity management and disability prevention for lymphatic filariasis in Luangwa district, Zambia: A qualitative study”

Title is now more in line with the content of the manuscript.

 Abstract, line 36 ; numerical values < than 10 may be better expressed in words; 4-four (circled)

Line 26; repetition of word ‘interview ‘? 

Line 73; Author Summary “costs of accessing healthcare services” is it actual costs or “perceived costs…? 

Line 88; Background “ 72 endemic countries…..” the number is much less with countries reaching the target of elimination. Isn’t this the estimate at the initiation of the GPELF?

Line 106; The target has been pushed to 2030

Line 114; ……..across all ten provinces [ ref]. Include a reference 

Line 119; …..places huge emphasis on

Lines 121-126 This sentence is too long, needs to be rephrased.

Line 150; ……which began in…./… begun in …

Lines 152 &153; What are the challenges faced in delivering the basic MMDP package for lymphoedema? 

Regarding “Very few health facilities are suitably equipped to provide basic MMDP package…. “ Basic lymphoedema management (which is irreversible if it progress) does not require sophisticated equipment …Needs clean water soap, basin, antibiotic medication, bandages etc. On the other hand hydrocele requires surgery (need equipment) but the condition can be reversed by surgery. 

Line 172; ……“ F campaign”… expand

Regarding Data collection; specify who did the interviews

Line 200;….. Health facility in-charges; Is it staff in-charge of the health facility

Line 218; Participants were invited….

Line 224 Sentence is incomplete 

Long distance to health facility- does it fall under approachability?

Similarly Lack of awareness- availability and accommodation?

Line 366; CHW ?

Lines 455-457; needs to be re-phrased as meaning is unclear.

Lines 461;…….LF patients……

Line 476; ….predominantly…. 

Conclusions

According to the study outcomes an urgent requirement is the establishment of a properly planned health education program in order to dispel false beliefs on disease causation and more importantly the correct measures that need to implement to prevent disease progression in the case of lymphedema. 

Line 502; these are” perceived causes” “…….causes perceived to be associated with the disease”

Line 510; HE should address issues regarding availability of care free-of-charge, fear of side effects etc

Line 514; “empower and motivate Lf patients and community to adhere to skin hygiene based self-care-regime” …?

The conclusion of the study is incomplete, suggest expansion focusing on deficiencies and how to rectify these?

Reviewer #2: The authors addressed all my comments and the manuscripts has significantly improved with focus on the qualitative aspect. I have the following few comments.

• Table 1 Summary of qualitative coding tree, would be better if it is presented on an actual tree how showing how each of the sub-themes relate one another. How is the perception affecting the service uptake in addition to the clear direct barriers such as cost and service availability? 

• Line 224 the statement ‘framework analysis approach was’ is incomplete. 

• Line 20 and 21: These authors contributed equally to this work is not clear, why the authors wanted to have two of them?

Reviewer #3: (No Response)

**Summary and General Comments**

Reviewer #1: As a preliminary study in an area where MMPD services are not properly established, the manuscript does provide insight as to community perceptions on disease,preferred care practices, reasons for their LF care behaviors etc. which may be useful for those designing MMDP activities. 

The qualitative study design does not provide information on the magnitude of each of the said perceptions, health seeking behaviors etc. This is a weakness that may be rectified by more formative studies as mentioned by the authors

Reviewer #2: This study used qualitative study and a range of study participants to understand the health beliefs and health seeking behaviour for LF MMDP services among communities living in Luangwa District, Zambia. Morbidity alleviation is one of the two pillars of The Global Programme for Elimination of LF. Often this aspect of the program is forgotten. The current study underlines the need for exploring patient perceptions and barriers to access for MMDP. This study is important for designing interventions tailored to the needs of the patients and programme which addresses patients concerns.

Reviewer #3: Overall, much improved. Just small edits to the abbreviations throughout need to be addressed e.g. CHW, MDA, LF sometimes written in full, sometime not

PLOS authors have the option to publish the peer review history of their article (what does this mean?). If published, this will include your full peer review and any attached files.

Reviewer #1: No

Reviewer #2: No

Reviewer #3: No
---

## [Decision Letter · Decision Letter 2]

15 Dec 2020

Dear Ms Maritim,

We are pleased to inform you that your manuscript 'Health beliefs and health seeking behavior towards lymphatic filariasis morbidity management and disability prevention services in Luangwa District, Zambia: Community and provider perspectives.' has been provisionally accepted for publication in PLOS Neglected Tropical Diseases.

Best regards,

Samuel Wanji

Guest Editor

Francesca Tamarozzi

Deputy Editor

Reviewer's Responses to Questions

**Key Review Criteria Required for Acceptance?**

**Methods**

-Are the objectives of the study clearly articulated with a clear testable hypothesis stated?

-Is the study design appropriate to address the stated objectives?

-Is the population clearly described and appropriate for the hypothesis being tested?

-Is the sample size sufficient to ensure adequate power to address the hypothesis being tested?

-Were correct statistical analysis used to support conclusions?

-Are there concerns about ethical or regulatory requirements being met?

Reviewer #2: (No Response)

**Results**

-Does the analysis presented match the analysis plan?

-Are the results clearly and completely presented?

-Are the figures (Tables, Images) of sufficient quality for clarity?

Reviewer #2: (No Response)

**Conclusions**

-Are the conclusions supported by the data presented?

-Are the limitations of analysis clearly described?

-Do the authors discuss how these data can be helpful to advance our understanding of the topic under study?

-Is public health relevance addressed?

Reviewer #2: (No Response)

**Editorial and Data Presentation Modifications?**

Reviewer #2: (No Response)

**Summary and General Comments**

Reviewer #2: The manuscript improved significantly. The authors have addressed all my comment. I do not have additional comments on the manuscript.

PLOS authors have the option to publish the peer review history of their article (what does this mean?). If published, this will include your full peer review and any attached files.

Reviewer #2: No

---

## [Editor Report · Acceptance letter]

18 Feb 2021

Dear Ms Maritim,

We are delighted to inform you that your manuscript, "Health beliefs and health seeking behavior towards lymphatic filariasis morbidity management and disability prevention services in Luangwa District, Zambia: Community and provider perspectives.," has been formally accepted for publication in PLOS Neglected Tropical Diseases.

Best regards,

Shaden Kamhawi

co-Editor-in-Chief

Paul Brindley

co-Editor-in-Chief
